# Acute Physiological and Perceptual Responses to Six Body-Weight Squat Exercise Variations

**DOI:** 10.3390/s25072018

**Published:** 2025-03-23

**Authors:** Daniel Santarém, Andreia Teixeira, António Amaral, Jaime Sampaio, Catarina Abrantes

**Affiliations:** 1Research Center in Sports Sciences, Health Sciences and Human Development, CIDESD, 5000-801 Vila Real, Portugal; danielrs@utad.pt (D.S.); abrantes@utad.pt (C.A.); 2Department of Sports Science, Exercise and Health, School of Life Sciences and Environment, University of Trás-os-Montes and Alto Douro, 5000-801 Vila Real, Portugal; andreiat@utad.pt (A.T.); al60620@alunos.utad.pt (A.A.)

**Keywords:** heart rate, muscle oxygen saturation, near-infrared spectroscopy sensor, strength training, training monitoring

## Abstract

Adequate exercise prescription requires a deep understanding of the body’s response to exercise. This study explored the responses of heart rate (HR), muscle oxygen saturation (SmO_2_), and perceived exertion (RPE) during six body-weight squat exercise variations. A total of 15 recreationally active participants (age: 28.2 ± 8.0 years; body mass: 71.1 ± 11.2 kg; height: 1.73 ± 0.08 m) were recruited. Six body-weight squat variations (deep, jumping, single-leg, uneven, unstable, and wall-sit) were randomly performed for 90 s. Results revealed that the jumping squat promoted a higher average and peak HR (165.3 ± 14.5 and 146.1 ± 14.8 bpm, respectively), and a lower average SmO_2_ and higher deoxygenation SmO_2_ in the soleus muscle (40.3 ± 15.4 and 46.0 ± 11.4%, accordingly). No differences were observed in recovery time or in the same SmO_2_ derived-parameters in the vastus lateralis muscle. The jumping variation promoted a greater response at a physiological level, both centrally, related to cardiovascular response, and peripherally, related to soleus SmO_2_. It was also the more demanding variation at both the overall and lower limb muscular level of RPE. This holistic view allows a precise identification of the response patterns in body-weight squat exercise variations to an acute session, with a training intervention providing additional information.

## 1. Introduction

The importance of real-time monitoring with high-precision technological resources in the sports context is unquestionable, with the main objective of maximizing performance and reducing the probability of non-functional overreaching, illness, and injury [1]. Advances in the technology of portable and wearable sensors allow for better dose–response identification, distinguishing between movement sensors (e.g., pedometer, accelerometer/gyroscope, and global positioning satellite [GPS]) and physiological sensors (e.g., heart rate monitor, temperature monitor, and integrated sensors) [2,3].

Physical performance and safety based on external load and internal load can be ensured by measuring objective (e.g., heart rate [HR], muscle oxygen saturation [SmO_2_], and subjective (e.g., perceived exertion [RPE]) parameters during and after exercise [4,5]. With HR being a standardized key indicator of acute physiological adaptation and the intensity of exercise presenting a linear relationship with maximum oxygen uptake over a large range of submaximal intensities [6], SmO_2_ is assessed non-invasively by measuring the amount of oxygenated and deoxygenated blood in the muscles using near-infrared spectroscopy sensors (NIRS) and it reflects the dynamic balance between oxygen supply and oxygen consumption [7]. In addition to HR, which is presented as a global objective measure, and SmO_2_, a specific objective measure, the RPE is characterized as a subjective, self-performed rating of the intensity of exercise, and can be evaluated through the 15-point RPE (Borg RPE scale 6–20) which is considered a valid and inexpensive tool for monitoring exercise intensity [8].

With exercise being considered a factor that induces several beneficial physiological responses, the application of differentiated stimuli appears to be advantageous in practical contexts. This process emerges through better adaptability to different constraints by stressing the neuromuscular system, thus becoming essential in forcing the body to adapt to new stimuli movements [9]. Due to their applicability and time-efficient alternative to traditional resistance exercise, body-weight exercises such as squats, push-ups, pull-ups, and sit-ups are widely used as resistance training exercises, and can be used as an effective strategy to improve muscular adaptations [10]. The squat is a functional weight-bearing exercise characterized as a multi-joint exercise and is one of the most widely applied in both the sports dimension, as it comprises biomechanical and neuromuscular aspects similar to most movements, and the quality of life dimension, with a good transfer to everyday functional movements [11]. The traditional squat is performed bilaterally; however, the conditions for performing the exercise may vary depending on the objectives to be achieved, with the most common being changes in terms of stance width, foot placement angle, and hip depth [12]. When considering various conditions of the squat (e.g., unstable surface, unilateral performance, uneven surface, different hip depth, ballistic movement, and different muscle contraction type), a more comprehensive view could be established of the different mechanisms involved in its implementation. Performing the single-limb squat optimizes neuromuscular control of the knee by increasing the coactivation of the posterior thigh muscles (i.e., hamstrings) [13]. In a recent systematic review by Neto et al. [14], it was noted that the manipulation of the contact surfaces would provide completely different muscle activation, with the application of exercises with uneven surfaces (e.g., step-up, lateral step-up, diagonal step-up, and crossover step-up) promoting very high muscle activation of the gluteus maximus (>100% maximal voluntary isometric contraction). Within the hip depth in the squat, the application of different depths, such as partial squat, parallel squat, and full squat, also seem to promote different muscle activations with the gluteus maximus being the only one (compared to the vastus medialis, vastus lateralis and biceps femoris) to have greater activation with increasing hip depth [15]. In a brief review, Baker [16] described the importance that the ballistic movement in the squat, through the application of the jumping squat, can have in planning the different phases of the load progression. Concerning muscle contraction type, Lee et al. [17] found that squatting using isometric contractions (wall squat) promotes less muscle activation in the rectus femoris, vastus lateralis, vastus medialis, biceps femoris, semitendinosus and semimembranosus muscles compared to dynamic contractions (general squat and Spanish squat).

Thus, the aim of this study was to compare the responses of HR, SmO_2_, and RPE to six body-weight squat exercise variations. It was hypothesized that the responses of HR, SmO_2_, and RPE would be different between variations, and that the variations that induced a higher HR and RPE would also elicit a more differentiated response from SmO_2_.

## 2. Materials and Methods

### 2.1. Study Design

A repeated-measures approach was used to examine differences in HR-, SmO_2_-, and RPE-derived parameters when performing six variations of the body-weight squat exercise. Participants were asked to (i) not perform moderate-to-vigorous intensity physical activity in the 24 h prior to the sessions; (ii) replicate their dietary and fluid intake for 24 h before the sessions; and (iii) abstain from alcohol, tobacco, and beverages that contained caffeine 3 h before participation [18]. The experimental design comprised two visits: the first session was the study presentation and exercise familiarization and the second session was the experimental session. The sessions were performed in a sports physiology laboratory under similar controlled conditions and separated by 48–72 h, with all testing procedures performed by the same researchers.

### 2.2. Participants

A total of 15 recreationally active participants were recruited from the academic community (Table 1). The estimated sample size was 15 participants, calculated for a repeated measures analysis of variance (ANOVA) between factors using G*Power (Version 3.1.9.7 Institut für Experimentelle Psychologie, Düsseldorf, Germany) for an effect size of 0.25, an α of 0.05, and a power of 0.7 (1–β). A pre-selection of potential participants was conducted to ensure that their prior experience with resistance training, specifically squat exercises, was aligned with the requirements of the study design. Participants were required to (i) be physically active by meeting World Health Organization recommendations [19] and (ii) be familiarized with the adequate technique for performing the squat exercises, with a minimum of one year’s experience in resistance training. Participants were excluded if they (i) had cardiovascular, pulmonary, or metabolic diseases; (ii) had more than two cardiovascular risk factors (e.g., age, smoking habits, obesity, dyslipidemia, a family history of heart attack, bypass surgery, or sudden death, sedentary lifestyle, hypertension, or pre-diabetes); and (iii) had skinfold thickness higher than 15 mm and had a dark skin tone (interference with NIRS sensor measurements). Participants showed no restriction to exercise based on their responses to the Physical Activity Readiness Questionnaire. This study was approved by the Ethics Committee of the University of Trás-os-Montes e Alto Douro (Doc28-CE-UTAD-2022), based at Vila Real (Portugal), and conducted according to the Declaration of Helsinki. All participants were informed in writing and orally about the study procedures and provided their written informed consent.

### 2.3. Procedures

In the first session, participants were instructed on the information related to the study and signed the informed consent, anthropometric measurements were taken, and a familiarization with the protocol procedures was performed. Body mass was evaluated using the SECA 701 balance (SECA Corporation, Hamburg, Germany), considering a tolerance limit of 0.2 kg. Height was measured with the SECA 220 stadiometer, coupled to the above-mentioned scale. Both were measured without shoes and light clothing. In addition, the skinfold thickness at these sites was assessed (Slim Guide, Minnesota, EUA) to ensure that the values were lower than 15 mm [20]. The skinfold thickness measurements were taken on the dominant side of the body with the participant in the lying position. Duplicate measures were made at each site and a third if the difference between duplicate measurements was higher than 2 mm [21]. The familiarization with the protocol procedures included the execution of the exercises, considering the technique, cadence, and duration of the exercise, and familiarization with the instruments to be used, namely the HR chest strap, NIRS sensors, and the RPE scale. During the familiarization with the RPE, standardized instructions and anchoring [22] were used to increase the validity and realibility of the subjective intensity characterization, by applying the memory anchor method [23]. In the same session, the placement location of the NIRS sensors in the analyzed muscle groups was determined, and participants were asked to keep the skin marks until the second session.

The second session was the experimental session, and the protocol procedures are represented in Figure 1.

Initially, the NIRS sensors were placed over the respective points, after confirmation of location, as well as the HR chest strap. Participants remained at rest for 5 min to obtain baseline values of each variable before the warm-up. The warm-up was standardized and comprised a general and a specific component: 12 shoulder, wrist, hip, and ankle rotations, 12 lateral trunk flexions, 6 adduction and abduction of the lower limbs, 2 inchworm with one leg out in front, the lateral rotation of the trunk and arm in extension (one repetition for each side), 6 forward lunges, 6 side lunges, and 6 traditional squats. After the warm-up, a 5 min rest period followed to ensure a physiological calibration of the different physiological variables. In addition, before each variation the participants remained in the standing position for 30 s, for the variables to stabilize. Then, the six body-weight squat exercise variations were randomly performed for 90 s each (controlling the muscle contraction time with cadence using a metronome), with 5 min passive rest in between. The variations were deep, jumping, single-leg, uneven, unstable, and wall-sit (Figure 2). These were described considering the following parameters: cadence, body position, foot position, hip angle, surface, and contraction type.

Deep squat: cadence—60 beats per minute (bpm); body position—trunk parallel to tibia, upper limbs follow the movement, at the end of the descending phase the glutes touch the tape, the chest must be kept up, and the knees pointing outward; foot position—shoulder-width apart; hip angle—110°; surface—stable; and contraction type—dynamic.

Jumping squat: cadence—40 bpm; body position—trunk parallel to tibia, upper limbs follow the movement, at the end of the descending phase the glutes touch the tape, the chest must be kept up, and the knees pointing outward; foot position—shoulder-width apart; hip angle—90°; surface—stable; and contraction type—dynamic.

Single-leg squat: cadence—40 bpm; body position—trunk parallel to tibia, upper limbs can hold a structure to help with balance, dominant leg remains on the ground; foot position—shoulder-width apart; hip angle—>90°; surface—stable; and contraction type—dynamic.

Uneven squat: cadence—60 bpm; body position—trunk parallel to tibia, upper limbs follow the movement, at the end of the descending phase the glutes touch the tape, the chest must be kept up, and the knees pointing outward; foot position—shoulder-width apart, and non-dominant foot on the platform (step); hip angle—90°; surface—stable; and contraction type—dynamic.

Unstable squat: cadence—60 bpm; body position—trunk parallel to tibia, upper limbs follow the movement, at the end of the descending phase the glutes touch the tape, the chest must be kept up, and the knees pointing outward; foot position—shoulder-width apart; hip angle—90°; surface—unstable; and contraction type—dynamic.

Wall-sit squat: cadence—not applicable; body position—lower back firmly against the wall, upper limbs relaxed at the side of the body; foot position—shoulder-width apart; hip angle—90°; surface—stable; and contraction type—isometric.

All exercises were visually monitored, and verbal instructions were transmitted to ensure proper technique.

### 2.4. Measurements and Instruments

SmO_2_ and HR data were visualized in real time via ANT+ connection to a personal computer using SPro software (RealTrack Systems, Almería, Spain). In addition, an inertial device WIMU PRO (RealTrack Systems, Almería, Spain) was used to synchronize the data from the different instruments.

#### 2.4.1. Heart Rate (HR)

HR was continuously recorded using a Garmin HRM3-SS HR chest strap (Garmin, Olathe, KS, USA) placed directly on the skin below the sternum, tight enough to stay in the same position with different body movements. The HR monitor (Size: 62 × 34 × 11 mm; Weight: 45 g) comprises an operating temperature range from −5 °C to 50 °C, and functions in 2.4 GHz ANT+ wireless communications.

#### 2.4.2. Near-Infrared Spectroscopy Sensor (NIRS)

SmO_2_ data were collected continuously using a validated and reliable [20] portable wireless NIRS sensor (Moxy Monitor, Fortiori Design LLC, Hutchinson, MN, USA). Based on modified Beer–Lambert law, the Moxy Monitor sensor (Size: 62 × 52 × 15 mm; Weight: 48 g) estimates SmO_2_ and total hemoglobin (tHb) levels in muscle capillaries below its point of position and consists of a light source transmitter positioned at 12.5 and 25.0 mm from the two detectors. In default mode, four wavelengths over 80 times every two seconds for an averaged output rate of 0.5 Hz are applied [20]. The sensor works by sequentially applying four wavelengths, at 680, 720, 760, and 800 nm, to assess the absorbance of oxyhemoglobin (O_2_Hb), deoxyhemoglobin (HHb), oxymyoglobin (O_2_Mb), and deoxymyoglobin (HMb). Using the law mentioned above, SmO_2_ is determined in percentage terms using the following Formula (1):(1)$$SmO_{2}=\frac{Hb+O_{2}Mb}{\left(O_{2}Hb+O_{2}Mb\right)+(HHb+HMb)}$$

Two NIRS sensors were placed on the skin over the muscles on the dominant side of the participants, in the vastus lateralis (VL) and soleus (SL) muscles, according to the recommendations of the SENIAM project for electromyography measurements [24]: in VL, at 2/3 on the line from the anterior superior iliac spinae to the lateral side of the patella, and in SL, at 2/3 of the line between the medial condyle of the femur to the medial malleolus. The locations were marked with a permanent marker to place them at the same position during the experimental session. The sensors were placed parallel to the muscle fibers. These were secured with an elastic therapeutic tape (Kinesio tape) and covered by a light shield suggested by the manufacturer, to mitigate probe movement and minimize light intrusion, respectively.

#### 2.4.3. HR and SmO_2_ Parameters Assessment

The maximum HR value reached during exercise was defined as HR_peak_, and the average HR (HR_avg_) was determined as the average value of the total exercise duration. The baseline SmO_2_ value (SmO_2baseline_) was defined as the average of the last 20 s preceding the exercise. The average exercise SmO_2_ (SmO_2avg_) was determined during the exercise period. The amplitude of deoxygenation (SmO_2deoxy_) was determined by the difference between SmO_2baseline_ and SmO_2min_. The time from SmO_2min_ to SmO_2baseline_ was defined as *t* SmO_2reoxy_.

The determination of HR- and SmO_2_-derived parameters is represented (Figure 3).

#### 2.4.4. Perceived Exertion (RPE)

The 6–20-point RPE scale [25] was used to measure the level of physical strain or perceived exertion. A rating of 6 is considered no exertion (i.e., rest), and a rating of 20 is considered maximal exertion (i.e., the most stressful exercise experienced). Participants were asked to report both their overall rating of perceived exertion (RPE-O) and rating of perceived exertion for the active muscles/lower limbs (RPE-AM).

### 2.5. Statistical Analysis

Data are presented as mean ± standard deviation, confidence interval (CI), and the *p*-value. All data were found to be normally distributed using the Shapiro–Wilk test. A repeated measures ANOVA with the Bonferroni post-hoc correction was conducted to examine the effect of exercise variations during exercise (on physiological and subjective variables) and during the recovery phase (on physiological variables). The sphericity of the data was assessed by Mauchly’s test. If the data violated the sphericity hypothesis, the Greenhouse–Geisser correction was applied. Effect sizes are presented as partial eta-squared (η_p_^2^) and were classified as small, moderate, and large, with values of ≤0.2, between 0.21, and 0.8, and >0.8, respectively [26]. Data were analyzed using the free statistic package JASP (version 0.17.1, Amsterdam, Netherlands). The level of significance was set at *p* ≤ 0.05.

## 3. Results

### 3.1. Physiological Responses

Physiological responses of HR_peak_, HR_avg_, SmO_2avg_, SmO_2deoxy_, and *t* SmO_2reoxy_ were compared between the six body-weight squat exercise variations and results are displayed in Figure 4.

There was a significant main effect of HR_peak_ on exercise variations, *F*(5.00, 55.00) = 7.66, *p* < 0.001, η_p_^2^ = 0.410. HR_peak_ was higher in the jumping squat than in the deep squat (mean difference = 18.2 bpm, 95% CI [10.1, 26.3], *p* < 0.001), the wall-sit squat (mean difference = 19.6 bpm, 95% CI [3.3, 35.9], *p* = 0.014), the uneven squat (mean difference = 24.1 bpm, 95% CI [9.4, 38.7], *p* < 0.001), and the unstable squat (mean difference = 14.1 bpm, 95% CI [2.2, 26.0], *p* = 0.015). The HR_avg_ was also significantly different between exercise variations, *F*(5.00, 50.00) = 7.07, *p* < 0.001, η_p_^2^ = 0.414. A higher response was exhibited in the jumping squat compared to the deep squat (mean difference = 13.1 bpm, 95% CI [4.1, 22.0], *p* = 0.003), wall-sit squat (mean difference = 15.3 bpm, 95% CI [2.5, 28.0], *p* = 0.015), uneven squat (mean difference = 19.1 bpm, 95% CI [7.5, 30.7], *p* = 0.010), and unstable squat (mean difference = 10.4 bpm, 95% CI [0.5, 20.3], *p* = 0.037).

The SmO_2avg_ in the SL muscle was significantly different between exercise variations, *F*(5.00, 55.00) = 10.05, *p* < 0.001, η_p_^2^ = 0.477. The jumping squat had lower values than the deep squat (mean difference = −23.5%, 95% CI [−42.1, −4.8], *p* = 0.010), uneven squat (mean difference = −20.9%, 95% CI [−38.3, −3.5], *p* = 0.014), and single-leg squat (mean difference = −22.0%, 95% CI [−41.0, −2.9], *p* = 0.019). In addition, the unstable squat presented lower values than the deep squat (mean difference = −11.4%, 95% CI [−19.7, −3.0], *p* = 0.005). There was no significant effect for SmO_2avg_ in the VL muscle. A significant main effect of SmO_2deoxy_ in the SL muscle was found, *F*(5.00, 50.00) = 19.56, *p* < 0.001, η_p_^2^ = 0.662. The jumping squat showed higher deoxygenation than the deep squat (mean difference = 27.4%, 95% CI [9.1, 45.7], *p* = 0.003), wall-sit squat (mean difference = 31.5%, 95% CI [19.6, 43.5], *p* < 0.001), uneven squat (mean difference = 26.8%, 95% CI [-13.8, 39.8], *p* < 0.001), and unstable squat (mean difference = 25.7%, 95% CI [12.4, 39.1], *p* < 0.001). Also, the single-leg squat had higher deoxygenation than the wall-sit squat (mean difference = 15.9, 95% CI [−28.8, −3.0], *p* = 0.012). There was no significant effect for SmO_2deoxy_ in the VL muscle. Both *t* SmO_2reoxy_ in the SL muscle and in the VL muscle showed non-statistically significant differences, thus exhibiting that the recovery time for the initial values is identical between the selected variations.

### 3.2. Perceptual Responses

The results of the exercise perceptual responses to RPE-O and RPE-AM are shown in Table 2.

There was a significant main effect of RPE-O on exercise variations, *F*(2.37, 16.61) = 5.71, *p* < 0.010, η_p_^2^ = 0.449. RPE-O for the jumping squat was significantly higher than for the deep squat (mean difference = −3.3 a.u., 95% CI [1.7, 4.8], *p* < 0.001), uneven squat (mean difference = 3.9 a.u., 95% CI [0.9, 6.9], *p* = 0.012), and unstable squat (mean difference = 3.5 a.u., 95% CI [0.3, 6.7], *p* = 0.030). Furthermore, a significant main effect of RPE-AM was also found, *F*(2.85, 28.53) = 12.20, *p* < 0.010, η_p_^2^ = 0.549. The jumping squat had higher RPE-AM values than the deep squat (mean difference = 5.3 a.u., 95% CI [2.2, 8.4], *p* = 0.001), uneven squat (mean difference = 4.6 a.u., 95% CI [1.7, 7.6], *p* = 0.002), single-leg squat (mean difference = 3.0 a.u., 95% CI [0.6, 5.4], *p* = 0.010), and unstable squat (mean difference = 5.0 a.u., 95% CI [2.9, 7.1], *p* < 0.001). Moreover, the wall-sit squat resulted in higher values than the deep squat (mean difference = 3.6 a.u., 95% CI [0.5, 6.8], *p* = 0.018).

## 4. Discussion

The main finding of this study was the specific response during exercise, with the jumping exercise causing a more pronounced output. That is, (a) HR showed a greater response in the jumping variation, suggesting a higher cardiovascular response; (b) SmO_2_ was more sensitive to changes in the SL muscle, indicating a larger gap between oxygen supply and oxygen consumption; and (c) the jumping variation also promoted higher values in RPE, thus generally agreeing with the hypothesis put forward.

### 4.1. Physiological Responses

The regulation of the circulatory system during exercise involves numerous complex biological processes as a result of the skeletal muscle’s extraordinary ability to adapt to variations in energetic needs. Knowing that the intrinsic activity of the sinus node, which is influenced by the parasympathetic and sympathetic nervous systems controls HR response, an HR increase at the start of exercise is mostly a result of parasympathetic withdrawal [27]. This increase in HR during exercise is due to a decrease in vagal tone followed by an increase in sympathetic flow and an increase in circulating catecholamine levels [28,29]. While HR_peak_ is related to the maximum effort exerted by the heart at a given moment, HR_avg_ is described as the effort exerted by the heart during exercise, with the combination of both variables providing important information on the cardiovascular response to exercise. The jumping squat variation was the one that showed significantly higher values compared to the others, and this response may be associated with the greater muscle mass required, i.e., a greater demand for oxygen from the skeletal muscle and thus the need to pump a greater amount of blood through the heart to meet the required needs, the greater demand for eccentric contractions, and the higher the variation in the center of mass [29]. Furthermore, considering the substantial involvement of aerobic energy metabolism in the squat exercise [30], examining responses in oxygen consumption during both exercise and recovery could add valuable information about the metabolic demand of these squat variations. Despite the similar conditions in terms of duration between variations, the use of different cadences and the different muscle contraction amplitudes characteristic of each variation can elicit a different sympathetic activity and elevated catecholamines (epinephrine and norepinephrine).

NIRS sensors evaluate SmO_2_ within the skeletal muscle microvasculature, reflecting the dynamic balance between oxygen supply to the muscle and oxygen consumption [7]. This variable is influenced by factors such as contraction velocity and intensity, where a longer contraction time and a higher intensity induces a higher SmO_2deoxy_ [31]; muscle fascicle length and fascicle angle, where lower SmO_2_ values are observed in the distal portion of the muscle compared to the proximal portion [32]; muscle fiber type, with SmO_2_ having a more accentuated decrease in type I fibers, when VL muscle was compared to the rectus femoris muscle [33]; and muscle contraction type, with lower SmO_2_ values being observed in dynamic contractions compared to isometric ones [34]. Most of the studies that analyzed the SmO_2_ response used the squat exercise with an external load and focused on just one muscle, VL [35,36]. The SmO_2avg_ in the VL muscle showed no significant differences when exercise variations were compared. This suggests that this muscle has an identical behavior in exercise at the level of the average response in the selected variations, and the addition of extra assessments such as electromyography could probably add relevant information. By assessing motor unit recruitment, the use of electromyography would make it possible to understand whether the changes in SmO_2_ behavior are due to increased muscle activation (resulting in a greater use of oxygen) or reduced oxygen supply (due to intramuscular pressure), providing deeper insight into the neuromuscular demands of different squat variations. Previous studies have shown that when the load was equalized between the back squat and front squat, no significant differences were observed in the VL muscle either, indicating that the energy demands were similar between these two exercises [35]. On the other hand, SmO_2avg_ in the SL muscle showed differences, with the jumping variation having lower average values. This response could be due to both decreased oxygen supply and/or increased oxygen consumption by the muscle because of the increased intramuscular pressure during repetitive contractions. One possible reason for different SmO_2_ responses could be related to the changes in blood flow, which are influenced by variations in the level of muscle recruitment [37], promoting increased blood flow for the more active muscles [38]. The SmO_2deoxy_, also referred to as the amplitude of muscle oxygen deoxygenation, is closely related to the SmO_2avg_ and may be associated with improved performance in athletes in other sports [39,40]. Although this variable showed an identical response between variations with no significant differences in the VL muscle, in the SL muscle, the jumping variation showed a greater amplitude compared to the others. This greater deoxygenation can induce greater muscle peripheral adaptations, more specifically at the mitochondrial and vascular level [39,40], making this an important response to be analyzed in conjunction with both the average behavior of SmO_2_ and HR. With the phosphocreatine system being related to SmO_2_ during exercise, both at the level of depletion and re-synthesis [41], the calculation of reoxygenation is an aspect that has gained a lot of emphasis due to the possibility of optimally determining the exercise–rest cycle [39,42,43], since it can represent the capacity for capillary perfusion. Neither the VL muscle nor the SL muscle showed differences in recovery time to baseline, showing that although the SmO_2_ response to exercise is more pronounced in the jumping variation, the recovery time is relatively identical to the other variations.

### 4.2. Perceptual Responses

The perception of exercise intensity provides one of the major intrinsic inputs for making decisions about the energy output [23]. With the development of various scales over the last few decades to find the most suitable for each context, the 6–20-point RPE scale has good validity in resistance exercise and is widely used [22]. Furthermore, within this scale it is possible to obtain two variables: RPE-O, which refers to the overall effort perceived throughout the body, and which is easily interpretable by most individuals, and RPE-AM [44], which refers to the muscle groups with the highest intervention, which in this case are the lower limbs. Since the jumping variation elicited the highest RPE, multiple underlying factors may contribute to the mechanisms driving this response. Given that concentric muscle actions generally elicit higher RPE than eccentric actions [45,46], due to the greater percentage of absolute maximal force required [47], a plausible explanation for the jumping variation having higher RPE values may be related to the need to produce more absolute force in the concentric muscle action to perform the jumping movement. Another reason for both RPE-O and RPE-AM to have shown higher values in the jumping variation is related to the influence that the muscle mass required has on the subjective perception of effort [48], and as a consequence, the greater use of muscle mass refers to higher RPE values [49]. This is because with the increase in muscle mass elicited, the number of corollary commands received by the sensory cortex also increases, intensifying the perception of the intensity of the effort [50]. Since participants used their arms to assist in impulsion in this exercise variation, this increase in upper body muscle mass could also be associated with a higher RPE. Although it was not the subject of this study and knowing that the types of muscle contraction can also influence the subjective response to effort, the different cadences may have induced slightly different concentric and eccentric muscle contraction times.

### 4.3. Study Limitations and Future Research

It is important to note that this study has some limitations that cannot be dismissed. First, the findings of this study are specific to recreationally active participants with previous experience in resistance training and squat exercises, thus the extrapolation of data must be done cautiously. Secondly, the external validity of these results may be affected by the statistical power set out in this study. Thirdly, although the exercise variations were performed randomly and the recovery time between exercises was sufficient for the variables in this study to achieve resting values, accumulated fatigue may have had some effect on the last exercise variations to be performed. The application of statistical methods such as an analysis of covariance or mixed-effects model can help attenuate the possible cumulative effect of fatigue.

Future studies should consider applying SmO_2_ measurements to several muscles to provide a deeper understanding of local internal load distribution in different muscle groups involved in exercise. Additionally, research should explore how potential variations in SmO_2_ may be linked to biomechanical factors, movement efficiency, and individual exercise technique adaptation.

## 5. Conclusions

The findings of this study show that certain variations have the same impact on the physiological and subjective studied variables, and that in most situations, the differences observed are related to the jumping variation. The selection of the jumping variation in a training context, compared to the others mentioned, promotes a greater response in all three dimensions. At a physiological level, it has a greater cardiovascular demand, analyzed through the HR response, and at a muscular level in the SL muscle, it has greater demand through the SmO_2_ response. In terms of the perceived exertion, it has a greater perception at both a general and specific level (RPE-O and RPE-AM, respectively). Monitoring the training load using a multidisciplinary approach makes it possible to analyze, identify, and select the most beneficial exercises, considering the desired objectives (cardiovascular effort; local muscular effort; and perceived exertion). The ecological validity of these findings is related to the possibility of applying a particular exercise variation depending on the specific context. By way of example, the jumping squat seems to be associated with greater cardiovascular and muscular demands and could be a viable option if the aim is to promote improvements in these physiological dimensions.

## Figures and Tables

**Figure 1 sensors-25-02018-f001:**
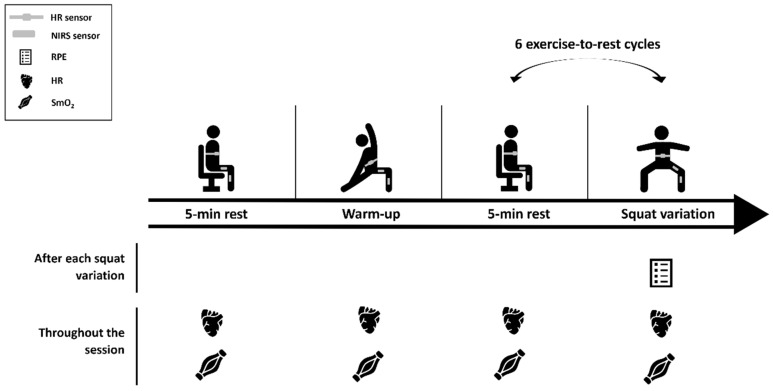
Schematic representation of the experimental session. HR, heart rate; NIRS, near-infrared spectroscopy sensor; RPE, rate of perceived exertion; and SmO_2_, muscle oxygen saturation.

**Figure 2 sensors-25-02018-f002:**
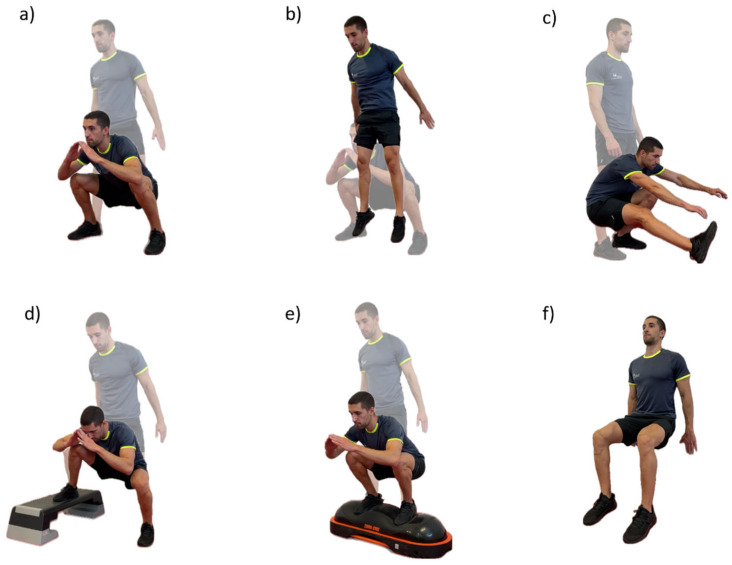
A representative example of the six body-weight squat exercise variations: (**a**) deep; (**b**) jumping; (**c**) single-leg; (**d**) uneven; (**e**) unstable; and (**f**) wall-sit.

**Figure 3 sensors-25-02018-f003:**
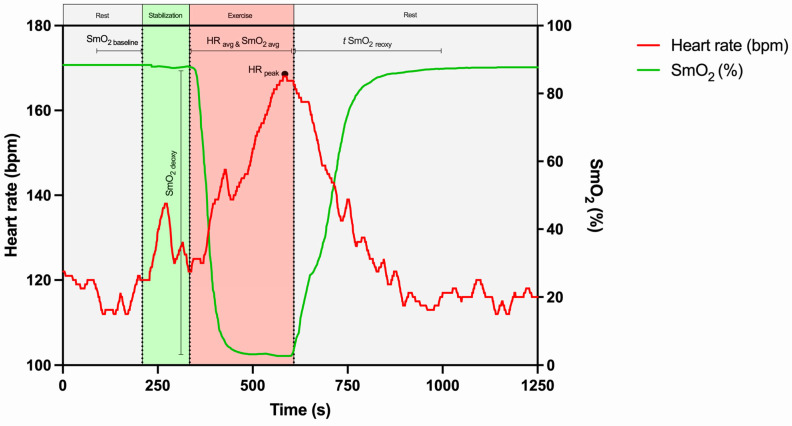
Representative example of SmO_2_- and HR-derived parameter calculations, based on a study participant. SmO_2avg_, average value of muscle oxygen saturation during exercise; SmO_2baseline_, average of the last 20 s preceding the exercise; SmO_2deoxy_, amplitude of muscle oxygen deoxygenation; *t* SmO_2reoxy_, time to recover to 100% of baseline muscle oxygen saturation; HR_avg_, average value of heart rate during exercise; HR_peak_, maximum value of heart rate during exercise.

**Figure 4 sensors-25-02018-f004:**
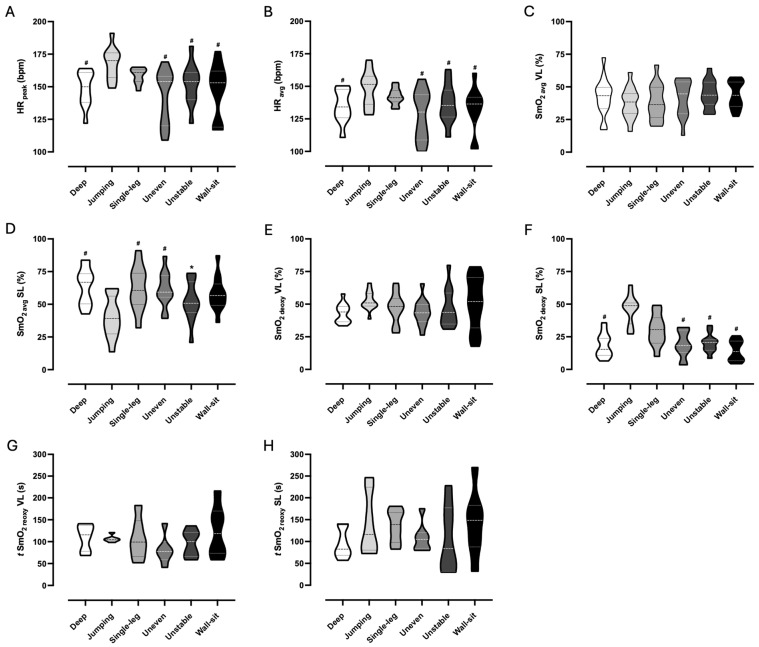
Violin plots represent a full distribution of raw data for HR_peak_ (**A**), HR_avg_ (**B**), SmO_2avg_ VL (**C**), SmO_2avg_ SL (**D**), SmO_2deoxy_ VL (**E**), SmO_2deoxy_ SL (**F**), *t* SmO_2reoxy_ VL (**G**), and *t* SmO_2reoxy_ SL (**H**) between the six body-weight squat exercise variations. In the violin plots, the dashed line indicates the median and dotted lines indicate the interquartile range (lower and upper lines). SL, soleus muscle; SmO_2avg_, average value of muscle oxygen saturation during exercise; SmO_2deoxy_, amplitude of muscle oxygen deoxygenation; *t* SmO_2reoxy_, time to recover to 100% of baseline muscle oxygen saturation; HR_avg_, average value of heart rate during exercise; HR_peak_, maximum value of heart rate during exercise; VL, vastus lateralis muscle. # statistical difference (*p* ≤ 0.05) to jumping squat. * statistical difference (*p* ≤ 0.05) to deep squat.

**Table 1 sensors-25-02018-t001:** Physical and physiological characteristics of the sample (n = 15).

Variable	Mean ± Standard Deviation
Age (years)	28.2 ± 8.0
Height (m)	1.73 ± 0.08
Body mass (kg)	71.1 ± 11.2
BMI (kg·m^−2^)	23.8 ± 2.8
VL skinfold (mm)	11.12 ± 3.97
SL skinfold (mm)	11.10 ± 3.94
HR rest (bpm)	75.0 ± 8.5
SmO_2_ rest in VL (%)	62.2 ± 14.3
SmO_2_ rest in SL (%)	54.5 ± 19.2

The values are presented as mean ± standard deviation. BMI, body mass index; HR, heart rate; SL, soleus muscle; SmO_2_, muscle oxygen saturation; VL, vastus lateralis muscle.

**Table 2 sensors-25-02018-t002:** Perceptual response to RPE-O and RPE-AM for six body-weight squat exercise variations.

Variables	Deep	Jumping	Single-Leg	Uneven	Unstable	Wall-Sit
**Perception**						
**RPE-O (a.u.)**	12.0 ± 0.8 ^#^	15.3 ± 0.9	12.3 ± 2.8	11.4 ± 1.8 ^#^	11.8 ± 1.8 ^#^	12.8 ± 2.8
**RPE-AM (a.u.)**	12.3 ± 1.7 ^#^	17.6 ± 1.7	15.9 ± 2.8 ^#^	12.9 ± 2.5 ^#^	14.6 ± 2.0 ^#^	12.6 ± 0.8 *

The values are presented as mean ± standard deviation. RPE-O, overall rating of perceived exertion; RPE-AM, rating of perceived exertion for the active muscles. ^#^ statistical difference (*p* ≤ 0.05) to jumping squat. * statistical difference (*p* ≤ 0.05) to deep squat.

## Data Availability

The raw data supporting the conclusions of this article will be made available by the authors on request.

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
