# Peer review of "Acute Physiological and Perceptual Responses to Six Body-Weight Squat Exercise Variations"

_sensors, 2025, doi:10.3390/s25072018_

Round 1

Reviewer 1 Report

Comments and Suggestions for Authors

The main idea of ​​this work was to compare the physiological and subjective response to different exercises. The authors found that the jumping load had the greatest change in cardiovascular variables. But the subjective perceptual response variable was not well described. In addition, the authors suggest that the types of muscle contractions may influence the subjective response to effort. However, this is not confirmed. Although, it is logical to assume that muscle contraction, which depends on the motor neuron nerve impulse, can be enhanced under the influence of mental activation. This can recruit an additional number of muscle units. I would like the authors to describe in more detail the identified mechanism of subjective stimulation of muscle contraction during jumping loads.

Reviewer 2 Report

Comments and Suggestions for Authors

Overall, authors clearly present their methodology and results. Especially, figures were quite helpful in communicating their ideas. Below are my comments and questions.

  1. Other than the parameters used in the study (HR, SmO2, RPE), what are other potential parameters than can be used for similar studies and how are the parameters used in this study better than the other potential parameters?
  2. I think a brief discussion on additional techniques or different measurements that could have provided even more insight would be a valuable addition to the manuscript.
  3. In part 2.1 where authors describe what they asked for the participants to do, are the times such as ’24 hours’, ‘3 hours’, ’48-72 hours’ considered standard in the field? If so, please add references. If those are the values that the authors chose, I wonder what the reasonings were.
  4. I understood the participant selection criteria but would like to know more about how exactly those 15 participants were selected (There must have been a lot more than 15 people who met the criteria). Also, among the general/wider population, how representative are those 15 participants?
  5. Do authors think any of the results/conclusion from the manuscript may change for different participants? (e.g. older people, overweight people, etc.)
  6. In line 106, authors set ‘a power of 0.7(1-beta)’. If I understand correctly, this means authors set the statistical power to be 70% (0.7). I read other work with higher statistical power such as 80%. Why did the authors choose 70%?
  7. In Fig 3, I suggested changing colors of y axes to match with the colors of the graphs (red and green) for improved readability.
  8. In Fig 3, can authors add a notation for SmO2_baseline as well?
  9. In Fig 3, according to the authors’ description, the unit of tSmO2_reoxy is ‘time’ but not ‘SmO2 (%)’. So I think the black vertical line that represents tSmO2_reoxy should actually be a horizontal line so that it can indicate ‘time’.
  10. For RPE analysis, I believe each participant would have different perceived exertion. Some might be more tolerant so only a few exercises are perceived RPE <10, while other participants are less tolerant and most of the RPEs were >10. Was there any normalization process to account for different perceptions for each participant? For example, individual correlation between RPE and HR or HR at RPE 12, or something similar could have been used to normalize participants’ response so that different participants’ responses could have been compared in the same scale. (ANOVA with Bonferroni post-hoc correction must have indirectly handled some of the normalization effect but I wonder if any more direct normalization efforts could have been made.)
  11. Could the order of 6 exercises have affected the results somehow? I understand that participants recovered to similar levels during rest period, but I am curious if accumulating fatigue, especially for 5th or 6th exercises, could have somehow affected the results. Follow-up question: If fatigue affected the results, how can it be processed for a fair comparison?
  12. More general question: How can the authors be sure that the results from each of the 6 exercises that were performed ‘in sequence’ be independent from each other? If 6 exercises were performed in a different order, do authors expect all the observations to be generally the same (e.g. pairwise comparisons authors claim to be significantly different or not significantly different)?
  13. In line 357, authors mention that most of the studies focus on VL but not on SL. Why is that the case? Should other studies also focus on SL?
  14. Can authors elaborate more on why they observed different behavior between VL and SL?
  15. Can authors elaborate more on the specific examples where the findings of this manuscript can be applied in real-world situations? (Maybe for example, designing effective exercise routine..?)
